# Underground Coal Mine Fingerprint Positioning Based on the MA-VAP Method

**DOI:** 10.3390/s20185401

**Published:** 2020-09-21

**Authors:** Mingzhi Song, Jiansheng Qian

**Affiliations:** School of Information & Electrical Engineering, China University of Mining and Technology, Xuzhou 221116, China; qianjsh@cumt.edu.cn

**Keywords:** fingerprint positioning, Wi-Fi, virtual access point, multi-association

## Abstract

The access points (APs) in a coal mine wireless local area network (WLAN) are generally sparsely distributed. It can, with difficulty, satisfy the basic requirements of the fingerprint positioning based on Wi-Fi. Currently, the effectiveness of positioning is ensured by deploying more APs in an underground tunnel, which significantly increases system cost. This problem can be solved by using the Virtual Access Point (VAP) method that introduces virtual access points (VAPs), which can be virtually arranged in any part of the positioning area without installing actual access points. The drawback of the VAP method is that the generated received signal strength (RSS) value of a VAP is calculated based on the mapping of RSS value from only one corresponding access point (AP). This drawback does not consider the correlation between different AP signals and the generated RSS value of a VAP, which makes the modeling of fingerprint samples and real-time RSS collection incomplete. This study proposed a Multi-Association Virtual Access Point (MA-VAP) method takes into account the influence of multi-association. The multi-association coefficient is calculated based on the correlation between the RSS values of a VAP and multiple access points (APs). Then, the RSS value generated by a VAP is calculated using the multi-association function. The real-time collected RSS values from multiple APs related to this VAP are the input of the multi-association function. The influence of the number of VAPs and their arrangement on positioning accuracy is also analyzed. The experimental positioning results show that the proposed MA-VAP method achieves better positioning performance than the VAP method for the same VAP arrangement. Combined with the Weight K-Nearest Neighbors (WKNN) algorithm and Kernel Principal Component Analysis (KPCA) algorithm, the positioning error of the MA-VAP method of the error distance cumulative distribution function (CDF) at 90% is 4.5 m (with WKNN) and 3.5 m (with KPCA) in the environment with non-line-of-sight (NLOS) interference, and the positioning accuracy is improved by 10% (with WKNN) and 22.2% (with KPCA) compared with the VAP method. The MA-VAP method not only effectively solves the fingerprint positioning problem when APs are sparse deployed, but also improves the positioning accuracy.

## 1. Introduction

Underground personnel positioning is crucial for coal mine safety management. At present, Radio Frequency Identification (RFID) [1], ZigBee [2] and Wi-Fi [3,4] are the main technologies for underground personnel positioning. Among them, the underground fingerprint positioning based on Wi-Fi has the widest application due to the low cost, great accessibility to many users, wide coverage, high transmission rate, and strong anti-interference ability.

Many important works in fingerprint positioning are mission-like noise reduction of fingerprint samples [5], the optimal selection of access points (APs) [6], and improvement of fingerprint positioning method [7,8]. However, there is little research on how to further improve the fingerprint positioning accuracy when APs are sparse deployed in coal mine tunnels.

The coal mine tunnels are usually long, and the underground wireless communication network considers only the communication coverage. Figure 1 shows APs deployment in underground tunnels of a coal mine in China’s Henan province, where most APs are sparsely deployed. The fingerprint positioning system in the existing scene of this coal mine cannot achieve accurate positioning, because it is difficult to ensure that more than three access point (AP) signals are simultaneously received by the mobile device in positioning tunnels, and the fingerprint positioning requires containing more than three valid received signal strength (RSS) values in the real-time RSS data. To ensure the accuracy of fingerprint positioning, a large number of APs have to be deployed in underground tunnels if the positioning area is very large. However, as shown in Figure 2, the coal mine special AP device is different from the general Wi-Fi equipment. It must be an intrinsically safe product, i.e., dust-proof and explosion-proof. Deployment of more expensive AP devices in the whole mine tunnels is only used to achieve fingerprint positioning, which will inevitably lead to a substantial increase in cost and the waste of resources. Therefore, it is difficult to adopt the kind of large-scale deployment of AP devices in an actual positioning system development. A higher cost-effective solution should be considered.

Zhang and Ding proposed a novel fingerprinting using Channel State Information (CSI) [9]. It is based on the technology of Multiple-Input Multiple-Output and Orthogonal Frequency Division Multiplexing (MIMO-OFDM). The CSI can describe multi-path propagation and takes account into the distance of features in each fingerprint. The CSI method improves the distinguishing of fingerprints. To improve the fingerprint positioning accuracy when APs sparse distribution, Zhang and Ding improved the CSI method, which using only a single AP [10]. The improved CSI method uses a novel phase decomposition method to obtain the phase of multi-path by an AP and uses the decomposed phase as a fingerprint after the feature exaction by Principal Component Analysis (PCA). However, the extraction of CSI needs special hardware, such as Intel 5300 Internet card. The prerequisite for the successful operation of the improved CSI method is that each mobile device should be equipped with the Internet card. If the number of users in coal mine is large, it will be a huge investment. Arshra and Hur proposed a fingerprint positioning method based on AP coverage area [11], which can better solve the radio fingerprint positioning problem when APs are sparsely distributed in a small positioning area. The proposed method in [11] mainly uses the concept of uniqueness of a single AP coverage area and overlapping of multiple APs coverage areas. The distribution of overlapping coverage areas is used to distinguish whether the target node is in the coverage area of a single AP or in the overlapping coverage area of multiple APs. The location estimation is based on the relative position of the coverage areas of the target node. Magh did S. A. and Magh did H. S. proposed an indoor human tracking mechanism using integrated onboard smartphones [12]. This mechanism uses RSS measurement between the smartphone and APs which exist in the same vicinity. It combines the RSS measurement with uncertainty calculations from onboard dead-reckoning measurements using Extended-Kalman filter, which can provide seamless and low cost fingerprint positioning. Due the user’s walking track data being recorded by multi-sensor integration of smartphone, the demand for the number of APs in the positioning method of [12] is not so important. In indoor positioning with APs sparsely distributed, the proposed method in [12] can achieve better positioning accuracy. However, in the application of underground fingerprint positioning, it is necessary to improve the intrinsic safety of smartphones. Kim proposed a novel 3D indoor localization scheme using virtual access point (VAP) [13]. VAP is defined as a virtual machine running on a single physical AP with different service set identifier (SSID). The distances are estimated using the received SSID, and the position of target node is estimated using a trilateral positioning method. Roth and Martin proposed a graph-theoretic approach to VAP correlation [14]. This method considers that it can be virtually constructed to the AP on a certain location in the indoor environment through statistical model. The statistical model is derived from a convex programming with an additional reverse convex constraint. Based on previous work [13,14], a fingerprint positioning-based VAP method was proposed in [15]. The VAP method is mainly intended for indoor environments containing a small number of APs [16]. The principle of the VAP method is to replace some of the APs with virtual access points (VAPs) in the off-line phase and then calculate the regression coefficient using the linear regression model between these VAPs and the remaining APs. A one-on-one association mapping between a VAP and an AP is established. In the on-line phase, the real-time RSS value from a VAP is generated by the one-on-one association mapping. After that Kalman Filter (KF) [17] and Particle Filter (PF) [18] are used to correct the RSS value generated by the VAP. KF and PF are mainly used to solve the impact of the instability of RSS values collected in the online phase on the positioning accuracy [15]. The corrected RSS values and real-time RSS values from the remaining APs are further used to do the optimal fingerprint matching. Finally, the estimated location is calculated. In the line-of-sight (LOS) environment, the positioning accuracy of the VAP method is similar to that used of all physical APs. In addition, in the non-line-of-sight (NLOS) environment, the positioning accuracy of the VAP method is higher.

In VAP method, the correlation between a VAP and an AP is represented by the regression coefficient, which is a constant and calculated by linear regression. By adopting the combination of the KF and the PF, the shortcomings of using only either linear or non-linear calibration of RSS can be overcome. However, this combination does not solve the problems completely in the calculation of the regression coefficient. Namely, when RSS values from different APs are simultaneously collected at the same reference point (RP) without using VAPs, there is a certain correlation between the collected RSS values. This correlation is not considered completely in the one-on-one association mapping calculation.

To overcome the problem of single association in VAP method, this paper proposes a Multi-Association Virtual Access Point (MA-VAP) method to improve the calculation precision of relationships between a VAP and APs. In the MA-VAP method, RSS values collected from a VAP and APs at each RP are considered to be measured data in the calculation of relationships between a VAP and APs. The paper also analyzes the influence of the number and deployed location of VAPs on positioning accuracy. The results show that by using the proposed MA-VAP method, a reasonable number of VAPs for the underground fingerprint positioning can be determined. The results of the underground positioning experiments in the environment with NLOS interference show that the MA-VAP method improves positioning accuracy by 10% (combined with WKNN) and 22.2% (combined with KPCA) compared with the VAP method.

## 2. Underground Fingerprint Positioning System Based on MA-VAP Method

### 2.1. VAP Method

The fingerprint positioning consists of two phases, the off-line phase and the on-line phase. The off-line phase is mainly responsible for the sampling of the RSS values from APs at the reference points (RPs) and constructing the radio fingerprint database. In the on-line phase, the RSS values collected in real time are compared with the fingerprint samples in the database, and the estimated location of target node is calculated by the optimal fingerprint matching. The VAP method proposed in [15] uses common APs to collect the RSS data in the off-line phase. In the on-line phase, VAPs are used to replace the part of common APs. The replaced AP is known as temporary AP (T-AP). The regression coefficient of VAP is calculated using the linear regression model between all temporary APs (T-APs) and the remaining common APs. The equation of regression coefficient is [15]:(1)$$\omega_{\mathrm{reg}}=\frac{N\sum_{n=1}^{N}f_{{\mathrm{AP}}_{n}}f_{{T-\mathrm{AP}}_{m}}-\sum_{n=1}^{N}f_{{\mathrm{AP}}_{n}}\sum_{m=1}^{M}f_{{T-\mathrm{AP}}_{m}}}{N\sum_{n=1}^{N}{\left(f_{{\mathrm{AP}}_{n}}\right)}^{2}-{\left(\sum_{m=1}^{M}f_{{T-\mathrm{AP}}_{m}}\right)}^{2}}$$
where $f_{{\mathrm{AP}}_{n}}$ is the RSS value of the *n*-th AP signal at a specific location; $f_{{T-\mathrm{AP}}_{m}}$ is the RSS value of the *m*-th T-AP signal at the same specific location; *N* and *M* are the number of APs and T-APs, respectively. The equation of one-on-one association mapping between a VAP and its corresponding AP is [15]:(2)$$S_{{\mathrm{VAP}}_{m}}=S_{{\mathrm{VAP}}_{0}}^{m}+\omega_{\mathrm{reg}}N$$
where $S_{{\mathrm{VAP}}_{m}}$ represents the RSS value generated by the *m*-th VAP in the on-line phase; $S_{{\mathrm{VAP}}_{0}}^{m}$ represents the simulated value of VAP when the distance between the *m*-th VAP and its corresponding AP is zero. Then, the PF is combined with the KF and used for linear and non-linear calibration of the generated RSS values of VAPs. 

The KF is used in the linear calculation of location estimation. The equation of the KF is [15]:(3)$$\left\{ \begin{array}{l} {\hat{x}}_{t|t-1}=F_{t}{\hat{x}}_{t-1|t-1}\\ P_{t|t-1}=F_{t}P_{t-1|t-1}F_{t}^{T}\\ G_{K}=H_{t}P_{t}H_{t}^{T}{\left(H_{t}P_{t}H_{t}^{T}+N_{t}\right)}^{-1}\\ z_{t}=H_{t}x_{t}+v_{t}\\ {\hat{x}}_{t}^{\mathrm{NEW}}=G_{K}^{\mathrm{NEW}}(z_{t}-H_{t}{\hat{x}}_{t})+{\hat{x}}_{t}\\ P_{t}^{\mathrm{NEW}}=P_{t}-G_{K}^{\mathrm{NEW}}H_{t}P_{t}\\ G_{K}^{\mathrm{NEW}}=P_{t}H_{t}^{T}(H_{t}P_{t}H_{t}^{T}+N_{t}) \end{array}\right.$$
where $x_{t}$ is the present location of the target node; ${\hat{x}}_{t|t-1}$ is the predicted state estimate of the target node from its RSS, with changes from time *t* − 1 to *t*; $z_{t}$ represents the observed measurement, with $H_{t}$ representing the matrix of it; $F_{t}$ represents the matrix of set predictions made from $x_{t-1}$ to $x_{t}$; $P_{t}$ is the predicted covariance matrix; $G_{K}$ is the Kalman gain matrix that outputs the minimum mean-square error by combining the variables from the predicted state and the observed measurement with noise covariance matrix $N_{t}$; ${\hat{x}}_{t}^{\mathrm{NEW}}$, $P_{t}^{\mathrm{NEW}}$ and $G_{K}^{\mathrm{NEW}}$ are the updated values of ${\hat{x}}_{t}$, $P_{t}$ and $G_{K}$, respectively. The updated value of ${\hat{x}}_{t}^{\mathrm{NEW}}$ will determine the latest or the last known location of the target node.

The PF is used to do non-linear calibration of the algorithm which is unlike the KF which is only capable of using the linear calibration. The equation of the PF is [15]:(4)$$\left\{ \begin{array}{l} P[x_{t}|z_{0:t}]=\sum_{i=1}^{N_{\mathrm{Par}}}\delta w_{t}^{i}(x)\\ w_{t}^{i}(x)=w_{t+1}^{i}(P[x_{t+1}|z_{t}]) \end{array}\right.$$
where $P[x_{t}|z_{0:t}]$ is the probability distribution of estimated location of target node, which will be the probability location of target node based on the RSS measurement with $N_{\mathrm{Par}}$ number of particles; $w_{t}^{i}(x)$ is the weight of a particle *i* at time *t*; δ is a normalized parameter. The weight $w_{t}^{i}(x)$ is normalized in order to obtain posterior density function.

In the scheme of the VAP method, the RSS value of a VAP is generated based on the RSS values of its corresponding AP. Since the data obtained by VAP can be used to predict RSS value of T-AP, the statistical analysis used by VAP method has advantages over using common APs [15,16]. However, $\omega_{\mathrm{reg}}$ in Equation (2) is a constant, $S_{{\mathrm{VAP}}_{m}}$ is calculated using the correlated generation of a single AP corresponding to ${\mathrm{VAP}}_{m}$. The calculation of $S_{{\mathrm{VAP}}_{m}}$ does not consider the interference between multiple APs in the actual positioning. The $S_{{\mathrm{VAP}}_{m}}$ calculated by Equation (2) probably contain some errors, which affect the positioning accuracy.

To verify the influence of simultaneous acquisition of multiple APs on the RSS value of an AP signal, experiment was carried out in the underground tunnel as shown in Figure 3. During the experiment, six APs were deployed in the order of ${\mathrm{AP}}_{1}{\sim \mathrm{AP}}_{6}$, the transmission range of AP was set at 45 m. Under the condition of different numbers of APs, a mobile device received 500 RSS values from ${\mathrm{AP}}_{1}$ at seven test locations (T1~T7). In the 802.11b/g Wi-Fi network, there are 13 channels, of which in theory, only the 1st, 6th and 11th channels do not interfere with each other [19]. Therefore, channel isolation between every two APs should be ensured as much as possible. ${\mathrm{AP}}_{1}$ and ${\mathrm{AP}}_{2}$ use 1st channel, ${\mathrm{AP}}_{3}$ and ${\mathrm{AP}}_{4}$ use 6th channel, ${\mathrm{AP}}_{5}$ and ${\mathrm{AP}}_{6}$ use 11th channel.

Figure 4 gives the RSS statistics from ${\mathrm{AP}}_{1}$ at different numbers of APs and different test locations in a tunnel. As shown in Figure 4, when the number of APs is more than three, the sample mean value of ${\mathrm{AP}}_{1}$ at each test location decreases, while the sample standard deviation increases. At test locations T4, T5, T6 and T7, especially, the changes of sample mean value and standard deviation are obvious. Because, the test locations of T4, T5, T6 and T7 are closer to ${\mathrm{AP}}_{2}$ than ${\mathrm{AP}}_{1}$. The interference of adjacent channels between ${\mathrm{AP}}_{1}$ and ${\mathrm{AP}}_{2}$ affected the RSS value received from ${\mathrm{AP}}_{1}$. In the VAP method, the calculation of RSS value generated by a VAP ignores the interference of adjacent channels during off-line sampling and on-line positioning. In the on-line phase, the optimal fingerprint matching between the undisturbed RSS generated values and the noisy fingerprint samples in the radio fingerprint database inevitably lead to the decline of matching accuracy. Moreover, if only one AP corresponding to a VAP is responsible for generating the RSS value of this VAP, KF and PF may not be able to accurately correct the generated RSS value of this VAP. Therefore, there is a modeling problem in the VAP method using one-on-one association mapping to calculate the generated RSS value of a VAP. The influence of multiple AP signals on the RSS generation of a VAP has to be considered. This influence can be determined by analyzing the statistical observed RSS values collected by APs and T-AP (VAP) in the off-line phase.

### 2.2. MA-VAP Method

The principle of the MA-VAP method is that the measured data of a large number of fingerprint samples are used to create the multi-association of RSS statistical distribution between a VAP and APs by least square fitting, and the generated RSS value of a VAP is calculated using the multi-association function. The main objective of the proposed MA-VAP method is to improve the solution of the multi-association coefficient.

In the off-line phase, *R* RPs are set in the positioning area. The RSS values from *N* APs and *M* T-APs are collected at each RP, where *T* represents the number of collections. The radio fingerprint database is recoded as:(5)$$\Phi =\Phi_{\mathrm{APs}}:\Phi_{T-\mathrm{AP}}=\left(\begin{array}{ccccc} f_{{\mathrm{AP}}_{1}}^{1} & \cdots & f_{{\mathrm{AP}}_{n}}^{1} & \cdots & f_{{\mathrm{AP}}_{N}}^{1}\\ \vdots & & \vdots & & \vdots \\ f_{{\mathrm{AP}}_{1}}^{r} & \cdots & f_{{\mathrm{AP}}_{n}}^{r} & \cdots & f_{{\mathrm{AP}}_{N}}^{r}\\ \vdots & & \vdots & & \vdots \\ f_{{\mathrm{AP}}_{1}}^{R} & \cdots & f_{{\mathrm{AP}}_{n}}^{R} & \cdots & f_{{\mathrm{AP}}_{N}}^{R} \end{array}\right)\;:\;\left(\begin{array}{ccccc} f_{{T-\mathrm{AP}}_{1}}^{1} & \cdots & f_{{T-\mathrm{AP}}_{m}}^{1} & \cdots & f_{{T-\mathrm{AP}}_{M}}^{1}\\ \vdots & & \vdots & & \vdots \\ f_{{T-\mathrm{AP}}_{1}}^{r} & \cdots & f_{{T-\mathrm{AP}}_{m}}^{r} & \cdots & f_{{T-\mathrm{AP}}_{M}}^{r}\\ \vdots & & \vdots & & \vdots \\ f_{{T-\mathrm{AP}}_{1}}^{R} & \cdots & f_{{T-\mathrm{AP}}_{m}}^{R} & \cdots & f_{{T-\mathrm{AP}}_{M}}^{R} \end{array}\right)$$ where $\Phi_{\mathrm{APs}}$ represents the matrix containing the fingerprint samples of all APs; $\Phi_{T-\mathrm{AP}}$ represents the matrix containing the fingerprint samples of all T-APs (VAPs); $f_{{\mathrm{AP}}_{n}}^{r}$ is the RSS data of the *n*-th AP signal collected at the *r*-th RP, recorded as $\left(f_{{\mathrm{AP}}_{n}}^{r}(1)\;\cdots \;f_{{\mathrm{AP}}_{n}}^{r}(t)\;\cdots \;f_{{\mathrm{AP}}_{n}}^{r}(T)\right)$; $f_{{T-\mathrm{AP}}_{m}}^{r}$ is the RSS data of the *m*-th T-AP signal collected at the *r*-th RP, recorded as $\left(f_{{T-\mathrm{AP}}_{m}}^{r}(1)\;\cdots \;f_{{T-\mathrm{AP}}_{m}}^{r}(t)\;\cdots \;f_{{T-\mathrm{AP}}_{m}}^{r}(T)\right)$. Thus, the matrix containing the *m*-th T-AP and all APs is:(6)$$\Phi_{{T-\mathrm{AP}}_{m}}^{\mathrm{APs}}=\Phi_{\mathrm{APs}}:\Phi_{T-\mathrm{AP}}^{m}=\left(\begin{array}{ccccc} f_{{\mathrm{AP}}_{1}}^{1} & \cdots & f_{{\mathrm{AP}}_{n}}^{1} & \cdots & f_{{\mathrm{AP}}_{N}}^{1}\\ \vdots & & \vdots & & \vdots \\ f_{{\mathrm{AP}}_{1}}^{r} & \cdots & f_{{\mathrm{AP}}_{n}}^{r} & \cdots & f_{{\mathrm{AP}}_{N}}^{r}\\ \vdots & & \vdots & & \vdots \\ f_{{\mathrm{AP}}_{1}}^{R} & \cdots & f_{{\mathrm{AP}}_{n}}^{R} & \cdots & f_{{\mathrm{AP}}_{N}}^{R} \end{array}\right)\;:\;\left(\begin{array}{c} f_{{T-\mathrm{AP}}_{m}}^{1}\\ \vdots \\ f_{{T-\mathrm{AP}}_{m}}^{r}\\ \vdots \\ f_{{T-\mathrm{AP}}_{m}}^{R} \end{array}\right)$$

Using the linear fitting of discrete fingerprint samples of the *m*-th T-AP and *N* APs, the correlation function of *m*-th T-AP and *N* APs is constructed [5], which is recorded as:(7)$$F(f;\gamma^{m})=\gamma_{1}^{m}f_{{\mathrm{AP}}_{1}}^{r}+\cdots +\gamma_{n}^{m}f_{{\mathrm{AP}}_{n}}^{r}+\cdots +\gamma_{N}^{m}f_{{\mathrm{AP}}_{N}}^{r}=\sum_{n=1}^{N}\gamma_{n}^{m}f_{{\mathrm{AP}}_{n}}^{r}=f_{{T-\mathrm{AP}}_{m}}^{r}$$
where r=1, ⋯, R. Using the RSS data in $f_{{\mathrm{AP}}_{n}}^{r}$ and $f_{{T-\mathrm{AP}}_{m}}^{r}$ as measured data, the correlation coefficient $\gamma_{n}^{m}$ in Equation (7) can be solved by least square method because of T≫N. In Equation (7), *r* corresponds to *R* RPs, correlation coefficients $\left(\gamma_{1}^{m}\;\cdots \;\gamma_{n}^{m}\;\cdots \;\gamma_{N}^{m}\right)$ correspond to different combination sequences at different RPs. Therefore, Equation (7) is extended to matrix form, which is recorded as:(8)$$F_{{T-\mathrm{AP}}_{m}}(\Phi_{\mathrm{APs}};\gamma^{m}(r))=\Phi_{\mathrm{AP}}\Delta_{\gamma^{m}(r)}=\Phi_{{T-\mathrm{AP}}_{m}}$$
where $\Delta_{\gamma^{m}(r)}$ is the column vector of the correlation coefficient γ of *m*-th T-AP, recorded as ${\left(\gamma_{1}^{m}(r)\;\cdots \;\gamma_{n}^{m}(r)\;\cdots \;\gamma_{N}^{m}(r)\right)}^{T}$. Since $\Delta_{\gamma^{m}(r)}$ changes with the value of *m*, the linear fitting of $\Phi_{\mathrm{APs}}$ and $\Delta_{\gamma^{m}(r)}$ is recorded as:(9)$$F_{{T-\mathrm{AP}}_{m}}(\Phi_{\mathrm{APs}};\omega_{r}^{m})=\Phi_{\mathrm{APs}}F_{\mathrm{APs}}(\Delta_{\gamma^{m}(r)};\omega_{r}^{m})=\sum_{n=1}^{N}\omega_{r}^{m}f_{{\mathrm{AP}}_{n}}^{r}\Delta_{\gamma (r)}=\Phi_{{T-\mathrm{AP}}_{m}}$$
where $F_{\mathrm{APs}}(\Delta_{\gamma^{m}(r)};\omega_{r}^{m})$ is the linkage function containing correlation coefficient vector with weight $\omega_{r}^{m}$. Using the fingerprint samples in $\Phi_{\mathrm{APs}}$ and $\Phi_{T-\mathrm{AP}}^{m}$ as measured data, the weight $\omega_{r}^{m}$ in Equation (7) can be solved by least square method because of T≫R in a small positioning area including *m*-th VAP. The Equation (9) establishes a multiple linear correlation between the fingerprint samples of a T-AP and multiple APs.

Through thee least square recursive solution of Equations (7) and (9), the equation of weight is:(10)$$\omega_{r}^{m}=a_{1}●\;(\frac{\min(f_{{\mathrm{AP}}_{n}}^{r})}{\max(f_{{\mathrm{AP}}_{m}}^{r})})/\left(\frac{\min(f_{{T-\mathrm{AP}}_{n}}^{r})}{\max(f_{{T-\mathrm{AP}}_{m}}^{r})}\right)+a_{2}●\;\frac{\mathrm{SD}(f_{{\mathrm{AP}}_{n}}^{r})}{\mathrm{SD}(f_{{T-\mathrm{AP}}_{m}}^{r})}$$
where $a_{1}$ and $a_{2}$ denote the correlation indexes, $0<a_{1}<a_{2}<1$,$a_{1}+a_{2}=1$; min(x) and max(x) represent the minimum and maximum values of collection *x*, respectively; SD(x) is the standard deviation. The solution of multi-association coefficient is:(11)$$\left\{ \begin{array}{l} \gamma_{n}^{m}=\frac{R\sum_{r=1}^{R}\left(\omega_{r}\bar{f_{{\mathrm{AP}}_{n}}^{r}}●\bar{f_{{T-\mathrm{AP}}_{m}}^{r}}\right)-\sum_{r=1}^{R}\bar{f_{{\mathrm{AP}}_{n}}^{r}}\sum_{r=1}^{R}\bar{f_{{T-\mathrm{AP}}_{m}}^{r}}}{R\sum_{r=1}^{R}{\left(\bar{f_{{\mathrm{AP}}_{n}}^{r}}\right)}^{2}-{\left(\sum_{r=1}^{R}\bar{f_{{T-\mathrm{AP}}_{m}}^{r}}\right)}^{2}}\\ \bar{f_{{\mathrm{AP}}_{n}}^{r}}=\frac{1}{T}\sum_{t=1}^{T}f_{{\mathrm{AP}}_{n}}^{r}(t)\\ \bar{f_{{T-\mathrm{AP}}_{m}}^{r}}=\frac{1}{T}\sum_{t=1}^{T}f_{{T-\mathrm{AP}}_{m}}^{r}(t) \end{array}\right.$$

In the on-line phase, all T-APs are removed and replaced by VAPs in their original locations. Suppose that the real-time RSS data received by target node is $\left(s_{1}\;\cdots \;s_{n}\;\cdots \;s_{N}\right)$. According to Equations (7), (10) and (11), the multi-association function of RSS generated by VAP can be obtained as follows:(12)$$\left\{ \begin{array}{l} s_{{\mathrm{VAP}}_{m}}=\delta_{{\mathrm{VAP}}_{m}}^{\mathrm{APs}}+\sum_{n=1}^{N}\gamma_{n}^{m}s_{n}\\ \delta_{{\mathrm{VAP}}_{m}}^{\mathrm{APs}}=S_{{\mathrm{VAP}}_{0}}^{m}●\frac{1}{N}\sum_{n=1}^{N}\gamma_{k} \end{array}\right.$$
where $\delta_{{\mathrm{VAP}}_{m}}^{\mathrm{APs}}$ is the calibration parameter, $S_{{\mathrm{VAP}}_{0}}^{m}$ is the same value as in Equation (1).

In Equations (5)–(9), the fingerprint samples of APs and T-APs in the radio fingerprint database are processed by multiple linear regression, which are used to deduce the multi-association function between APs and a T-AP. The Equations (10) and (11) are the correlation weights and correlation coefficients of each AP calculated by multiple linear regressions of observation data from APs and T-APs in the off-line phase. Then, the multi-association functions of all VAPs in the positioning area can be calculated using Equation (12). In the on-line phase, the target node receives the real-time RSS values from APs, and the RSS values of VAPs are generated according to Equation (12) and these real-time RSS values.

Finally, the RSS values generated by all VAPs are optimized by the combination of the KF and the PF [15,20]. The processes and effects of the KF and the PF are the same as that in the VAP method. The KF and the PF in the MA-VAP method use Equations (3) and (4) to correct the generated RSS values of VAPs, respectively.

### 2.3. System Structure and Operation Process

The underground fingerprint positioning system based on the MA-VAP method consisted of the AP devices, mobile devices, and positioning system software. The core of the positioning system includes the VAP generating module, KF filtering module, and PF filtering module. In the off-line phase, the VAP generating module calculates the multi-association coefficient of each VAP after collecting multiple sets of RSS values from all the APs and T-APs at each RP. In the on-line phase, the VAPs generates their RSS values for a target node based on the relationships with other APs (refer to Equation (12)). Then, the KF filtering module and the PF filtering module correct the obtained RSS values of the VAPs in turn. The influence of NLOS exists in many areas of underground tunnel, which makes the collected RSS value in real time unstable. The combination of KF and PF can effectively solve the influence of the fluctuation of collected RSS values on the fingerprint matching [15]. The workflow of the proposed positioning system is shown in Figure 5. Weight K-Nearest Neighbors (WKNN) algorithm [21] and Kernel Principal Component (KPCA) algorithm [22] are usually used to realize the optimal fingerprint matching for location estimation in Figure 5. The principle of WKNN algorithm is that K weighted fingerprint samples which are similar to real-time RSS data are selected, and the estimated location of target node is calculated from the geometric average of RPs’ coordinates corresponding to K fingerprint samples. 

## 3. VAP Arrangement

### 3.1. Problem Analysis

In general, using enough number of APs in the positioning area can effectively ensure positioning accuracy. The positioning accuracy of a certain area can be ensured using at least three APs in that area. However, in the radio fingerprint positioning of underground WLAN, the AP deployment tends to be sparse, as shown in Figure 1. The optimal coverage of the AP signal is the first consideration in underground wireless communication environments of almost all coal mines in China. To improve the quality of communication, the AP communication range should have some redundant coverage when the AP is deployed. Consider the general underground tunnel wireless communication network presented in Figure 6, the redundant coverage of the AP is represented by region R2. However, since only one AP signal can be received in regions R1 and R3, the basic requirement of fingerprint positioning for AP deployment is not satisfied. Although region R2 has better communication quality, it does not meet the requirement of fingerprint positioning for AP deployment. Thus, in order to meet the requirement for AP deployment in the whole area of Figure 6, the number of APs should be increased at least by two times, which will significantly increase the hardware cost for the case of a relatively long underground tunnel. Therefore, it is impractical to deploy a large number of APs in an underground tunnel. 

Using the MA-VAP method, a solution of the requirement for AP deployment in Figure 6 is show in Figure 7 [14,23]. In the off-line phase, ${T-\mathrm{AP}}_{1}$ and ${T-\mathrm{AP}}_{2}$ are added in underground tunnel. The relationships between the two T-APs and ${\mathrm{AP}}_{1}$ and ${\mathrm{AP}}_{2}$ are obtained by Equation (12), and the radio fingerprint database is generated after the RSS values collected. In the on-line phase, ${T-\mathrm{AP}}_{1}$ and ${T-\mathrm{AP}}_{2}$ are removed and replaced by ${\mathrm{VAP}}_{1}$ and ${\mathrm{VAP}}_{2}$, respectively. The target node obtains four RSS values during positioning, two of which are collected from ${\mathrm{AP}}_{1}$ and ${\mathrm{AP}}_{2}$, and the other two are obtained from the RSS values generated by ${\mathrm{VAP}}_{1}$ and ${\mathrm{VAP}}_{2}$. The RSS value of VAP is calculated by Equation (12). At this moment, no matter where the target node is in R1, R2, or R3, the RSS data of target node contains at least three RSS values from different data sources. This meets the requirement of fingerprint positioning for AP deployment. Because ${T-\mathrm{AP}}_{1}$ and ${T-\mathrm{AP}}_{2}$ are only used for off-line data acquisition and can be reused in other tunnel regions, the cost of hardware is not increased greatly in the actual fingerprint positioning application. Therefore, the MA-VAP method can realizes the fingerprint positioning in the underground WLAN without permanent deploying additional APs. 

It can be inferred from Figure 7 that the number of VAPs affects the final positioning accuracy. The VAP arrangement is significant for the MA-VAP method. The number of VAPs and their arrangement in the underground positioning area are analyzed experimentally in the following section.

### 3.2. VAP Arrangement Experiment

The AP device used in the experiments in this work was KJJ660W, it is shown in Figure 2. Its maximum transmission range was 100 m. The mobile device is shown in Figure 8, its operating system was Android 8.0. The positioning system software was developed by Spring4, Spring MVC, and Hibernate, and MySQL version 8.0.18 was used to create a database.

The experimental positioning environment is shown in Figure 9. As presented in Figure 9, there were three APs and 120RPs in the positioning tunnel [24]. For the purpose of easy viewing, the horizontal (**X**) and vertical (**Y**) coordinate display ratio in Figure 9b is 2.5:1. The transmission range of AP (KJJ660W) was set at 45 m. To verify the impact of the number of VAPs on positioning accuracy, seven VAPs were used and arranged, as shown in Figure 9b. Eight experiments were conducted and the number of the used VAPs increased gradually from zero to seven, the order of addition was ${\mathrm{VAP}}_{1}{\sim \mathrm{VAP}}_{7}$. In each experiment with different VAP numbers, the VAP method and the MA-VAP method were used as the positioning methods. In the off-line phase, seven T-APs were deployed at the locations of their corresponding VAPs, and 200 RSS samples were collected at each RP [25]. Then, the one-on-one association mapping in the VAP method and the multi-association function in the MA-VAP method of each VAP were calculated respectively. In the on-line phase, all T-APs were removed, and 200 positioning results using the VAP method were recorded for each of eight different numbers of VAPs. Similarly, 200 positioning results using the MA-VAP method were recorded for each of eight different numbers of VAPs. The RSS values of VAPs were obtained by the VAP method and the MA-VAP method respectively as a comparison of different algorithms. The RSS data received by the mobile device were corrected by the combination of the KF and the PF. Finally, the Weight K-Nearest Neighbors (WKNN) algorithm [21] was used to achieve optimal fingerprint matching.

The experiment results are shown in Figure 10 and Figure 11, and given in Table 1. When the positioning was conducted without using the VAPs, the positioning error was large, which was because when there were only three APs deployed, in most areas, three AP signals could not be received at the same time, so the positioning results included large errors. When the number of VAPs in the positioning area increased, and the VAPs were deployed in a reasonable manner, the positioning accuracy of the VAP method and the MA-VAP method increased significantly. When the number of VAPs was larger than four, the real-time RSS data received at nearly all locations contained more than three RSS values, including those received from APs and generated by VAPs. At this scenario using the MA-VAP method, the positioning accuracy improved compared with the results of without VAP or less than three VAPs. When the number of VAPs was larger than six, the positioning error of the error distance cumulative distribution function (CDF) at 90% of the VAP method and the MA-VAP method were 4 m and 3.5 m, respectively. Compared with the results of four VAPs and five VAPs, the positioning accuracy of using six VAPs or seven VAPs increased. 

To verify the performance of the MA-VAP method when VAPs are arranged in different locations under the same number of VAPs, experiments were carried out in three scenarios as shown in Figure 12. In this experiment, except for the different locations of VAPs, other parameters were the same as the previous experiment. In each experiment with different VAP-deployed locations, the VAP method and the MA-VAP method were used as the positioning methods. In the off-line phase of each scenario, four T-APs were deployed at the locations of their corresponding VAPs, and 200 RSS samples were collected at each RP. Then, the one-on-one association mapping in the VAP method and the multi-association function in the MA-VAP method of each VAP were calculated respectively. In the on-line phase, all T-APs were removed, and 500 positioning results were recorded for each scenario of each positioning method. In Scenario 1, the VAPs were arranged on both sides of the tunnel and in the middle area of two adjacent APs deployment locations. In Scenario 2, the VAPs were arranged on both sides of the tunnel and closer to the APs deployment locations. In Scenario 3, the VAPs were arranged on one side of the tunnel, and the distance between VAP and VAP or between VAP and AP was equal in the ***X***-axis direction. In all three scenarios, at least three RSS values were guaranteed to be received at each location.

The experiment results are shown in Figure 13 and given in Table 2. The positioning accuracy of the VAP method for Scenario 1 and Scenario 3 was similar, while the positioning accuracy of Scenario 2 was slightly worse than that of Scenario 1 and 3. Similarly, the positioning accuracy of the MA-VAP method of Scenario 2 was slightly lower than that of Scenario 1 and Scenario 3. Because the VAPs were arranged closer to the APs in Scenario 2, the region R1 and region R2 shown in Figure 12b had a certain distance from VAPs or APs nearby. In the off-line phase, the RSS values collected from RPs in R1 and R2 fluctuated greatly, which make the fingerprint samples of those RPs have larger noise. In the on-line phase, even if the RSS values generated by VAPs in R1 and R2 was relatively stable after KF and PF, there were some errors when doing the optimal fingerprint matching with fluctuated fingerprint samples of RPs in R1 and R2. Therefore, the positioning accuracy at R1 and R2 was not accurate enough. The VAPs arrangement in Scenario 1 and Scenario 3 was more reasonable for the tunnel with APs sparse distribution.

Therefore, the VAPs have to be reasonably deployed when the MA-VAP method is used for positioning in an underground tunnel with APs sparse distribution. For an underground tunnel with large extension range, the APs deployment in different areas is not the same, so the VAP arrangement scheme is also different. In the tunnel with APs sparse distribution, the important recommendation for the VAP arrangement is that no less than three real-time RSS values are received simultaneously at any location. For linear tunnel, the VAP arrangement location is usually in the middle area of two APs which are continuously deployed and far away from each other. At the corner of the tunnel, it depends on the APs deployment near the adjacent area of each tunnel branch. On the premise that more than three real-time RSS values can be received at the locations near the corner of the tunnel, the locations of VAPs should be evenly distributed in the areas with low AP deployment density. In addition, positioning accuracy will be improved if more VAPs are installed. However, positioning accuracy will not be infinite improvement. The number of VAPs should not be too large; otherwise, the positioning efficiency will be affected, and the unnecessary investment cost will be introduced.

## 4. Results and Discussion

### 4.1. Establishment of Underground Positioning Experiment

To verify the positioning performance improvement of the MA-VAP method in comparison to the VAP method, the positioning experiments were conducted in two different tunnel environments as shown in Figure 14a and Figure 15a respectively [26]. In experimental environment Ⅰ, the movement of miners and mine cars introduced NLOS interference to AP signal. In experimental environment Ⅱ, the movement of miners and tramcars similarly introduced NLOS interference to AP signal.

The plan of experimental environment I is shown in Figure 14b. Five APs were deployed in the positioning area. The number of RPs was 153 and the RPs were numbered from 1 to 153. To ensure that three or more RSS values (from APs and generated by VAPs) were received simultaneously at each location, eight T-APs were reasonably arranged in the positioning area. The transmission range of the AP (KJJ660W) was 45 m. Two hundred RSS samples were collected at each RP in the off-line phase. The RSS distribution diagram of APs and VAPs at each RPs is shown in Figure 16, the default value of RSS was −85 dBm. The sample mean values of eight VAPs in Figure 16 were collected by the eight corresponding T-APs in the off-line phase. The plan of experimental environmentⅡis shown in Figure 15b, the horizontal (**X**) and vertical (**Y**) coordinate display ratio is 2.5:1. Five APs were deployed in the positioning area. The number of RPs was 120 and the RPs were numbered from 1 to 120. To ensure that three or more RSS values were received simultaneously at each location, six T-APs were reasonably arranged in the positioning area. The transmission range of the AP (KJJ660W) was 45 m. Two hundred RSS samples were collected at each RP in the off-line phase. The RSS distribution diagram of APs and VAPs at each RPs is shown in Figure 17. The sample mean values of six VAPs in Figure 17 were collected by the six corresponding T-APs in the off-line phase. In both experimental environments, all fingerprint samples were collected when there were no miners or underground transport vehicles passed by, and the interference of other human factors was minimized.

In the on-line phase, there were two positioning scenarios. One was that all T-APs were used as the common APs (without VAPs), and the other was that all T-APs were replaced with VAPs. The first scenario was used to obtain a reference for the positioning results using all APs in the positioning area. The second scenario was used for realizing underground positioning adopting the VAP method and the MA-VAP methods. In the VAP method and the MA-VAP methods, the generated RSS data were corrected by the combination of the KF and the PF. KF and PF were mainly used to correct the unstable RSS values with NLOS interference. The WKNN algorithm and KPCA algorithm was used for optimal radio fingerprint matching in each scenario. The value of *K* in WKNN was four and Gaussian kernel function was adopted by KPCA. The different combinations of several positioning methods used in the experiments were given in Table 3. During the experiments in each experimental environment, 500 positioning results were recorded in each combination group. In addition, in experimental environment Ⅰ, some miners and mine cars passed near the test locations sometimes during positioning tests. In experimental environment II, all positioning tests were carried out with miners and tramcars passing near the test locations.

### 4.2. Experimental Results and Analysis

The positioning results of experimental environmentⅠare shown in Figure 18 and given in Table 4. The comparison results between group A and group C or group B and group D show that the VAP method with APs sparse distribution achieved the similar positioning accuracy as when the number of APs was sufficient (the first scenario without VAPs), due to the correction of the KF and the PF. The positioning errors of the VAP method combined with WKNN and the WKNN algorithm without VAPs at the 90% CDF were both 4.5 m. The positioning errors of the VAP method combined with KPCA at the 90% CDF was only 0.5 m less than that of the KPCA algorithm without VAPs. In the comparison of group C and group E, group D and group F, the MA-VAP method had higher positioning accuracy than the VAP method. The positioning errors of the MA-VAP method combined with WKNN and KPCA at the 90% CDF were 4 m and 3.5 m, respectively. The positioning accuracy was improved by 11.1% and 22.2% compared with the VAP method. Because the influence of multiple APs on the generated RSS value of VAP was taken into account in the calculation of multi-association coefficient. In this way, the problems of the constant regression coefficient and the one-on-one association mapping between VAP and AP corresponding to it in VAP method were solved. 

The positioning results of experimental environmentⅡare shown in Figure 19 and given in Table 5. In experimental environmentⅡ, the RSS values collected in real time were unstable due to the influence of NLOS interference. The positioning errors of WKNN and KPCA without VAPs at the 90% CDF were 6.5 m and 5.5 m. The positioning errors of the MA-VAP method combined with WKNN and KPCA at the 90% CDF were 4.5 m and 3.5 m, respectively. The positioning accuracy was improved by 10% and 22.2% compared with the VAP method. The comparison results between Figure 18 and Figure 19 show that the VAP method and MA-VAP method can also realize the fingerprint positioning when APs are sparsely distributed in NLOS interference environment. It gets benefit from the dynamic correction of RSS values by the KF and PF [15]. Consequently, the fingerprint positioning by the MA-VAP method can improves positioning accuracy without changing the number and deployment of existing APs.

## 5. Conclusions

In an underground wireless communication environment with APs sparse distribution, it is difficult to realize a fingerprint positioning system without making great changes to the existing underground AP deployment. The VAP method provides a better choice for the positioning method in the situation when the APs are sparsely distributed in a positioning area. Its basic principle is to arrange a certain number of VAPs in the positioning area. In the off-line phase, the correlation coefficient between a VAP and APs are established using the collected RSS data. The RSS value of the VAP is generated based on the regression coefficient. The regression coefficient effectively expands the valid data of RSS collections during the fingerprint positioning process. In addition, the KF and PF are used to correct the RSS value generated by the VAP. KF and PF effectively solve the problem of unstable real-time RSS value.

However, the VAP method treats the correlation between VAP and AP as a one-on-one association mapping. The RSS value generated by the association mapping of a VAP includes a certain error, and this error can hardly be filtered out even after the KF and PF corrections. Therefore, the positioning accuracy of VAP method can still be improved.

In this article, the correlation between a VAP and APs is determined based on the multi-association coefficient. The coefficient is calculated using the statistical data analysis of RSS values collected from VAPs and APs at each RP in the off-line phase. The multi-association coefficient is not a constant, which differs from the regression coefficient in the VAP method. When the RSS generated by a VAP is calculated, the real-time collected RSS values from multiple APs related to this VAP should be considered. Due to the low coverage density of APs in the existing underground tunnel wireless communication network, it can only guarantee the effective communication coverage of the whole area. This coverage density cannot realize the normal working of the fingerprint positioning system. Therefore, it is necessary to analyze how many VAPs should be arranged in the positioning area. The article analyzes the influence of the number and deployed location of VAPs on positioning accuracy. At least three signals from different sources should be included in the collection of real-time RSS data. Combined with the WKNN algorithm and the KPCA algorithm, the experimental results show that the positioning accuracy of the MA-VAP method is better than of the VAP method. In addition, the positioning accuracy of MA-VAP method is improved by 10% (with WKNN) and 22.2% (with KPCA) compared with the VAP method in the environment with NLOS interference.

This work focuses on developing a more efficient and low cost positioning system with higher positioning accuracy in the positioning scene with APs sparse distribution. The application of the MA-VAP method enables the construction of a Wi-Fi-based positioning system without changing the structure of the underground wireless communication network. Furthermore, when the underground positioning changed greatly and the RSS distributions of AP signals changed, the MA-VAP method has to resample the underground positioning areas and recalculate the multi-association functions of all VAPs, which will affect the normal work of the positioning system. The MA-VAP method has poor adaptability to the positioning environment change, which is the main problem to be solved in further research. In addition, further research will also concentrate on the validity correction of RSS values from APs, and these RSS values will be used to generate the RSS value of a VAP.

## Figures and Tables

**Figure 1 sensors-20-05401-f001:**
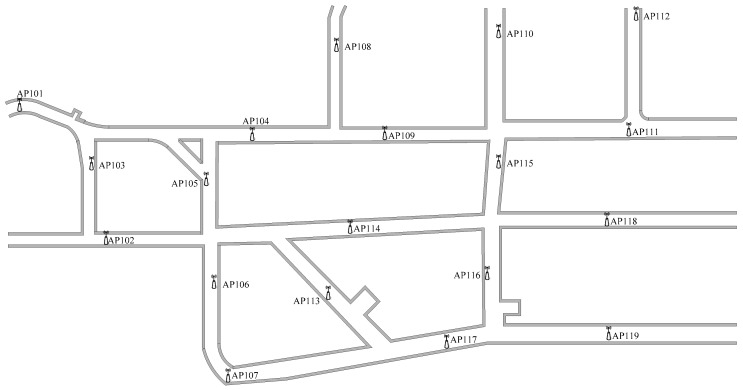
Diagram of underground AP deployment.

**Figure 2 sensors-20-05401-f002:**
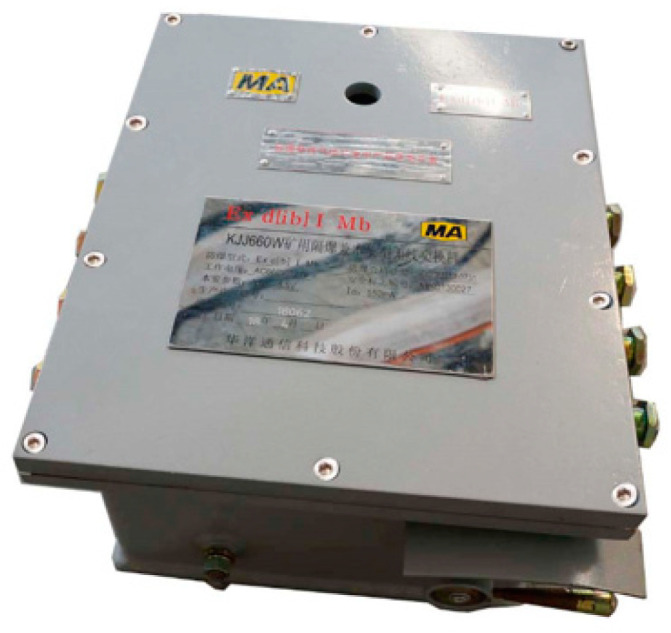
The image of the AP device KJJ660W used in the coal mine.

**Figure 3 sensors-20-05401-f003:**
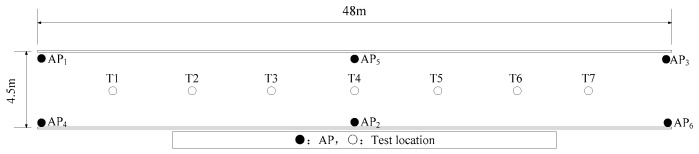
Plan of the undergrounds tunnel.

**Figure 4 sensors-20-05401-f004:**
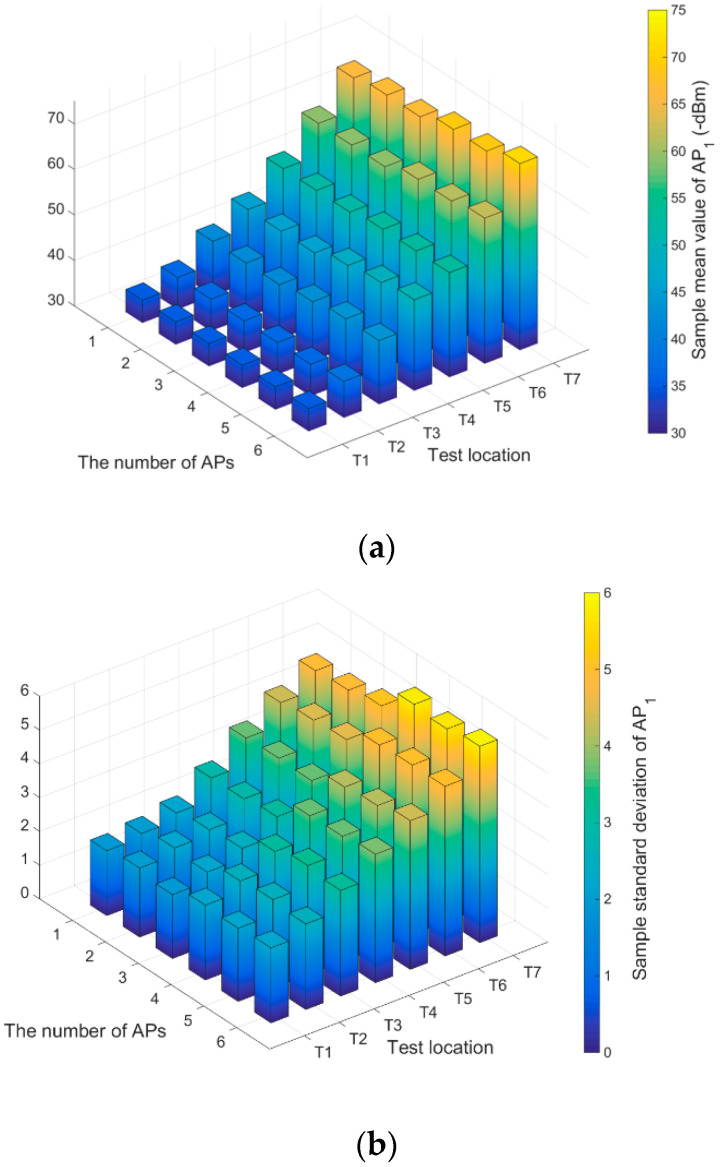
The RSS statistics from ${\mathrm{AP}}_{1}$ at different numbers of APs and different test locations in a tunnel: (**a**) Mean value of ${\mathrm{AP}}_{1}$; (**b**) Standard deviation of ${\mathrm{AP}}_{1}$.

**Figure 5 sensors-20-05401-f005:**
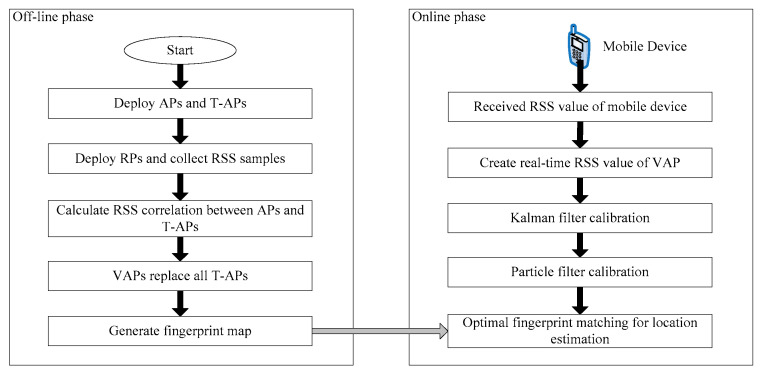
The workflow of the proposed positioning system algorithm mainly uses kernel function to change fingerprint samples into high-dimensional feature space, which makes the sample set with non-linear characteristics have linear separability. The process of optimal fingerprint matching includes feature extraction and feature vector matching. Compared with WKNN algorithm, KPCA algorithm has higher accuracy in fingerprint matching.

**Figure 6 sensors-20-05401-f006:**
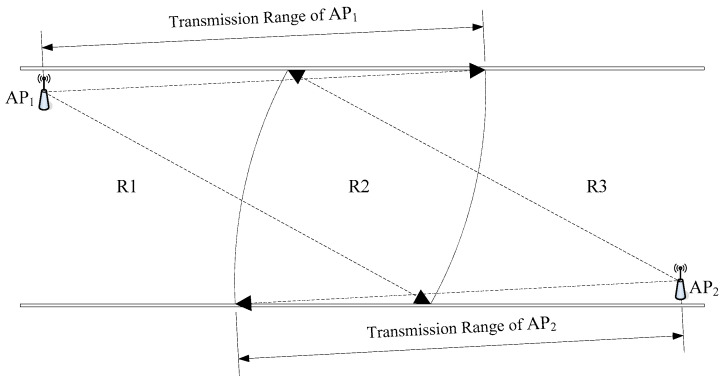
Diagram of a general underground tunnel wireless communication network.

**Figure 7 sensors-20-05401-f007:**
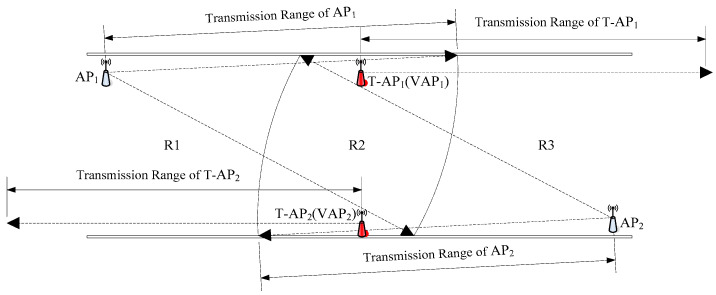
Diagram of VAP arrangement.

**Figure 8 sensors-20-05401-f008:**
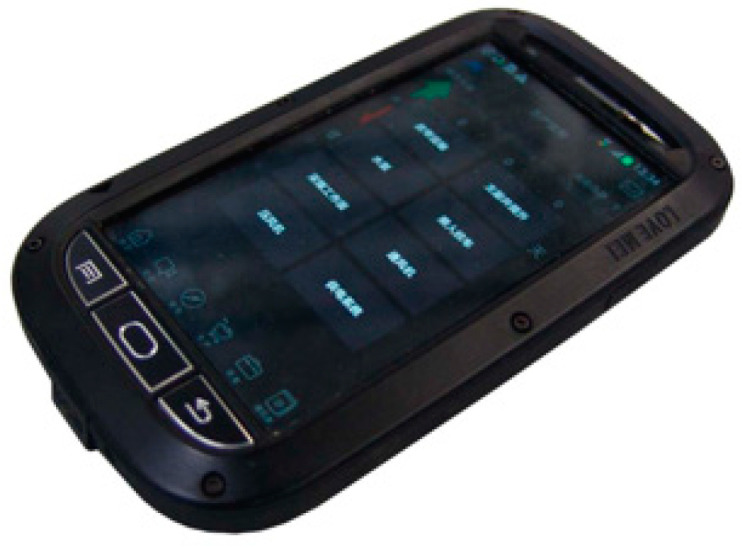
The intrinsic safety mobile phone.

**Figure 9 sensors-20-05401-f009:**
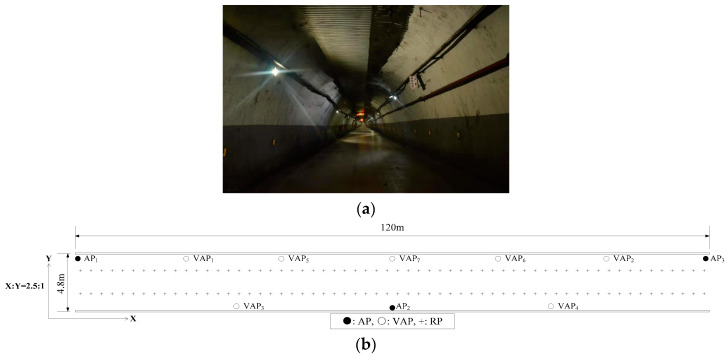
The experimental positioning scenario: (**a**) Picture of the underground tunnel; (**b**) Plan of the underground tunnel.

**Figure 10 sensors-20-05401-f010:**
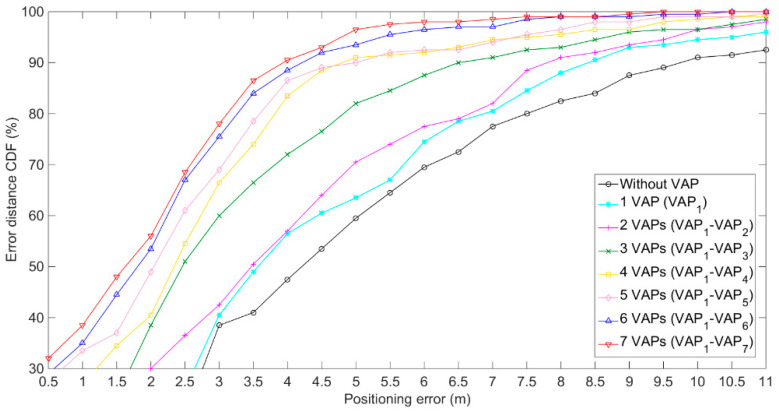
Positioning results of the VAP method for eight different numbers of VAPs.

**Figure 11 sensors-20-05401-f011:**
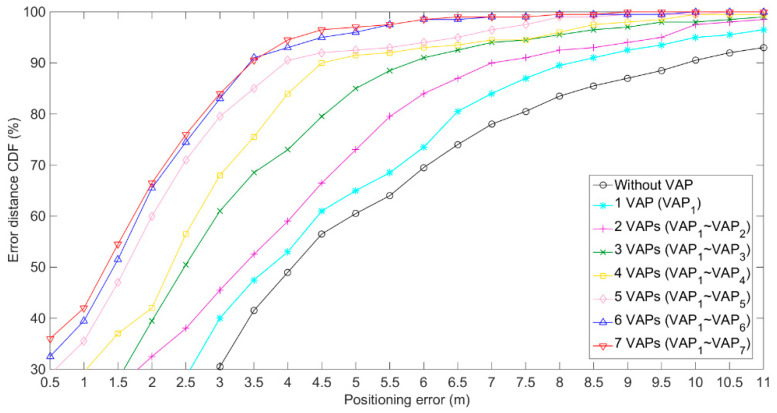
Positioning results of the MA-VAP method for eight different numbers of VAPs.

**Figure 12 sensors-20-05401-f012:**
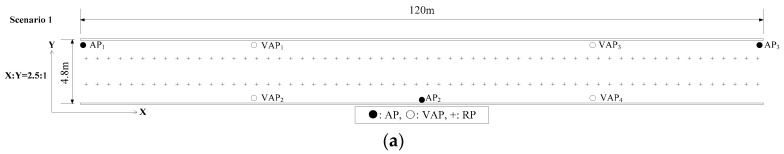
The experimental positioning scenarios using MA-VAP method under the same number and different locations of VAPs: (**a**) Scenario 1; (**b**) Scenario 2; (**c**) Scenario 3.

**Figure 13 sensors-20-05401-f013:**
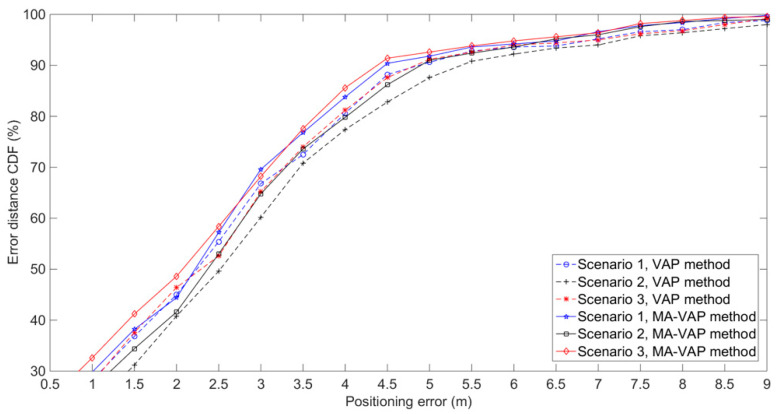
Positioning results of the VAP method and MA-VAP method for three different scenarios.

**Figure 14 sensors-20-05401-f014:**
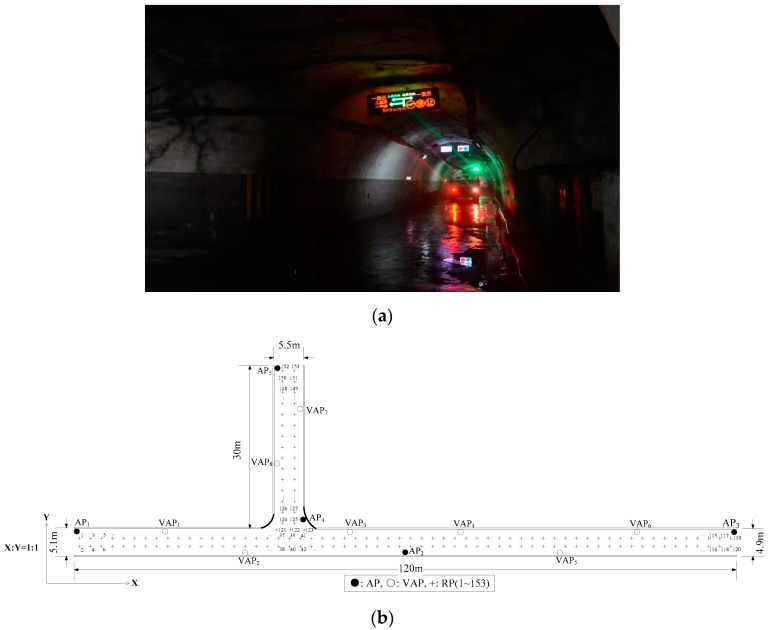
Diagram of experimental environment I: (**a**) Picture of the underground tunnel; (**b**) Plan of the underground tunnel.

**Figure 15 sensors-20-05401-f015:**
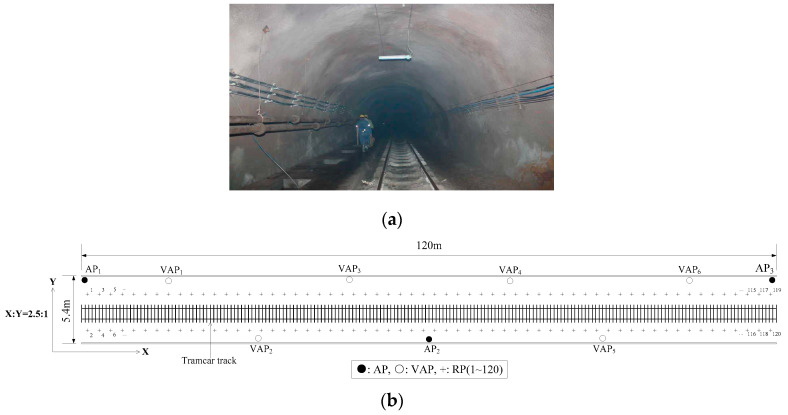
Diagram of experimental environment II: (**a**) Picture of the underground tunnel; (**b**) Plan of the underground tunnel.

**Figure 16 sensors-20-05401-f016:**
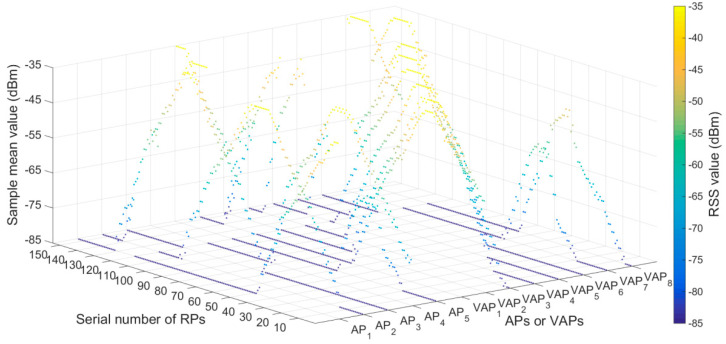
RSS distribution diagram of APs and VAPs at each RPs in experimental environment I.

**Figure 17 sensors-20-05401-f017:**
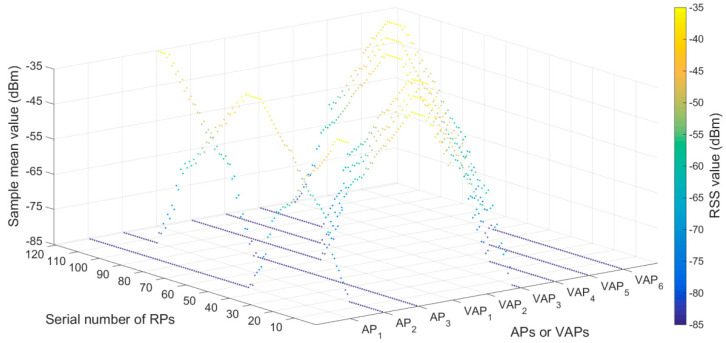
RSS distribution diagram of APs and VAPs at each RPs in experimental environment II.

**Figure 18 sensors-20-05401-f018:**
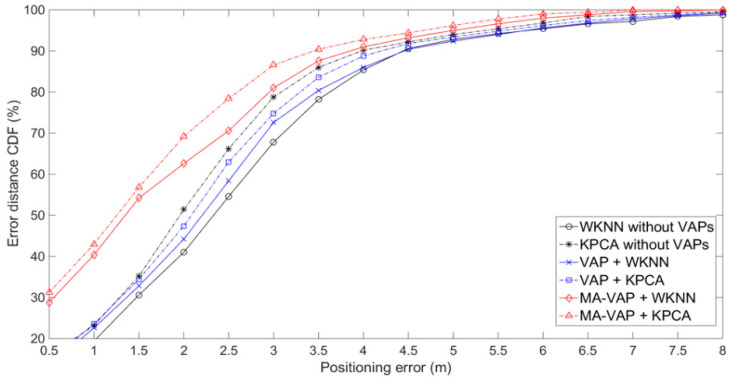
Comparison of the positioning results of different combination groups in experimental environment.

**Figure 19 sensors-20-05401-f019:**
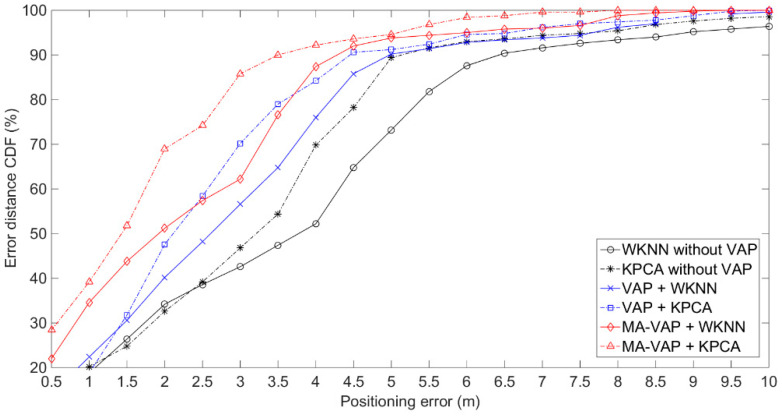
Comparison of the positioning results of different combination groups in experimental environment.

**Table 1 sensors-20-05401-t001:** Positioning errors of the VAP method and the MA-VAP method at the 90% CDF for eight different numbers of VAPs.

The Number of VAPs	Positioning Error (m)	
	VAP Method	MA-VAP Method
0	10.5	10.5
1 (${\mathrm{VAP}}_{1}$)	8.5	8.5
2 (${\mathrm{VAP}}_{1}{\sim \mathrm{VAP}}_{2}$)	8	7
3 (${\mathrm{VAP}}_{1}{\sim \mathrm{VAP}}_{3}$)	6.5	6
4 (${\mathrm{VAP}}_{1}{\sim \mathrm{VAP}}_{4}$)	5	4.5
5 (${\mathrm{VAP}}_{1}{\sim \mathrm{VAP}}_{5}$)	5	4
6 (${\mathrm{VAP}}_{1}{\sim \mathrm{VAP}}_{6}$)	4.5	3.5
7 (${\mathrm{VAP}}_{1}{\sim \mathrm{VAP}}_{7}$)	4	3.5

**Table 2 sensors-20-05401-t002:** Positioning errors of the VAP method and the MA-VAP method at the 90% CDF for three different scenarios.

Scenario	Positioning Error (m)	
	VAP Method	MA-VAP Method
1	5	4.5
2	5.5	5
3	5	4.5

**Table 3 sensors-20-05401-t003:** The different combinations of several positioning methods used in the experiments.

Combination Group	Positioning Methods
A	WKNN without VAPs
B	KPCA without VAPs
C	VAP + WKNN
D	VAP + KPCA
E	MA-VAP + WKNN
F	MA-VAP + KPCA

**Table 4 sensors-20-05401-t004:** Positioning errors comparison of different combination groups at the 90% CDF in experimental environment.

Combination Group	Positioning Error (m)
A	4.5
B	4
C	4.5
D	4.5
E	4
F	3.5

**Table 5 sensors-20-05401-t005:** Positioning errors comparison of different combination groups at the 90% CDF in experimental environment.

Combination Group	Positioning Error (m)
A	6.5
B	5.5
C	5
D	4.5
E	4.5
F	3.5

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
