# Peer review of "Underground Coal Mine Fingerprint Positioning Based on the MA-VAP Method"

_sensors, 2020, doi:10.3390/s20185401_

Round 1
Reviewer 1 Report
The authors have proposed the MV-VAP method to improve the positioning accuracy for the underground coal mine fingerprinting positioning scheme. The manuscript looks written well, nonetheless, the authors need to reflect the below comments to logically insist on excellence in the proposed method. 1. In the off-line phase, the authors have proposed the MV-VAP that generates the RSS on the virtual space from the measured RSS. Fig. 10 shows the results that the proposed method can improve the positioning accuracy than the method which is not used VAP. However, since the related works had been continuously studied, the authors should compare the proposed method with at least one of the algorithms using the VAP. 2. The proposed method uses a combination of PF and KF, but it is necessary to describe the physical meaning of each filter on the proposed method. In particular, Section 2.2 needs to describe what each equation physically means, rather than just deriving the MA-VAP method as an equation. 3. At line 473, ‘wit1h’ is wrong.
Reviewer 2 Report
This paper presents an approach to WiFi-based positioning with virtuall Access Points (AP) focusing on underground environments, namely coal mines.
The methods used for positioning are pretty standeard, while the focus is on methods used for establishing so-called Virtuall Access Points (VAP) among the sparsely located physical APs in the coal mine. The paper is properly structured and reads quite well, although the English still could be improved, it is awkward in some sentences (e.g. "anthropic factors").
The paper has its merits, and a piblishable research results, but there are still some issues (I understand this is a second round of review and the paper has been improved). Namely:
the related work is quite scarce and it should be extended with more recent nresults on indoor positioning with WiFi/smartphone focusing on methods that deal with the unavailability and sparseness of the WiFi signals also by multi-sensor integration,
- - the role of the (Extended ?) Kalman Filter and the Particle Filter in positioning has to be clarified. I see that there is some improvement on this topic (red text), but to a reader who did not read [13] it is largely unclear. Please, devote a paragraph to explain this with proper schemes and/or equations.
- - I had doubts as to citation of some related work, e.g. why [22] is cited in the experimental sections - does it have anything in common with hour dataset, as sugested by the text on page 9 ?
Round 2
Reviewer 1 Report
Before accept to publish this manuscript, there is one small concern. Since the authors only insisted on the advantages based on the simulation results, many readers may be thought about the drawback of the proposed method. Thus, if possible, please briefly describe the disadvantage of the proposed scheme both in Sections 4 and 5. After that, I suggest further research includes the direction of improvement to overcome the disadvantage.
Consequently, since the authors have reflected all the comments elaborately, I don't feel like needing more major comments.
Author Response
Please see the attachment.

This manuscript is a resubmission of an earlier submission. The following is a list of the peer review reports and author responses from that submission.
Round 1
Reviewer 1 Report
1. The experiment environment is too ideal, like in the walkway environment of a general building, or in a tunnel, rather than in a mine. The content does not match the title. 2. The description of the MA-VAP strategy is too cluttered, and it is difficult to find out where the main improvements compared to the VAP method, which difficulties have been overcome, and those innovative strategies 3. The experimental parameters are not clearly stated, and the experimental results reveal too little, such as the lack of AP and VAP signal strength distribution diagrams in the tunnel, which can easily be composed of the RSSI received by each RP. Consequently, it is difficult to repeat the experiment based on the paper, and it is difficult to infer the correctness of the result from the experimental parameters. In addition, there is no statistical metric called "confidence probability" (in the vertical axis of Figure 9). 4. For an academic paper, it is very important to be able to repeat the experiment and get similar results based on the experimental environment and data described. This paper does not satisfy this criteria.
Reviewer 2 Report
Page 3 - Figure 1:
Highlight the difference between Figure 1-(a) and Figure 1-(b)
Page 4 - the last paragraph: "Then, the KF filtering ...."
Check the reference correctly, since the authors refer to KF filtering module in [14] and [8]
Page 5 - Figure 2, spell check mobile device
Page 7 - the first paragraph: "The positioning experiments were ... "
Rephrase this paragraph to explain that 8 experiments were conducted and the number of the used VAPs was increased gradually from 0 to 7
It is important to explain why the VAPs were placed in their positions? Is there any recommendation for the VAPs distribution?
Page 8 - the middle of the last paragraph: "At the same time, eight ..."
You have to explain the reason for the arrangement

Reviewer 3 Report
Please see the attached file for comments.

Round 2
Reviewer 1 Report
The fatal shortcomings of this paper are that the experimental environment is too ideal and the assumptions of the working environment (coal mine) are too weird to meet the expectations of readers who read this paper because of the title of the paper.
I know that a modern mine will have the environment presented in the paper, but in this environment (as shown in Figure 9 (a) and Figure 13 (a)), the power supply will not be a problem. Even in Figure 13 (a), there is also a large LED display board. Therefore, it is very unusual that it is assumed in the paper that the erection of AP is difficult.
Second, I think there will be a lot of machine tools and equipment moving or working in a working environment in a mine, so the signal quality of Wi-Fi AP will not be maintained in a stable state, there will be a lot of interference, or even intermittent I’m looking forward to the author’s research on these conditions, rather than an experiment in an empty tunnel.
Reviewer 3 Report
See the attached file for comments.
